# A Panel Comprising Serum Amyloid A, White Blood Cells and Nihss for the Triage of Patients at Low Risk of Post-Stroke Infection

**DOI:** 10.3390/diagnostics11061070

**Published:** 2021-06-10

**Authors:** Leire Azurmendi Gil, Laura Krattinger-Turbatu, Juliane Schweizer, Mira Katan, Jean-Charles Sanchez

**Affiliations:** 1Human Protein Sciences Department, University of Geneva, 1211 Geneva, Switzerland; leire.azurmendi@unige.ch (L.A.G.); laura.krattinger@unige.ch (L.K.-T.); 2Department of Neurology and University of Zurich, University Hospital, 8057 Zürich, Switzerland; Juliane.Schweizer@usz.ch (J.S.); Mira.Katan@usz.ch (M.K.)

**Keywords:** stroke, infection, antibiotic therapy, panel of biomarkers, predictive algorithms

## Abstract

Accurate and early prediction of poststroke infections is important to improve antibiotic therapy guidance and/or to avoid unnecessary antibiotic treatment. We hypothesized that the combination of blood biomarkers with clinical parameters could help to optimize risk stratification during hospitalization. In this prospective observational study, blood samples of 283 ischemic stroke patients were collected at hospital admission within 72 h from symptom onset. Among the 283 included patients, 60 developed an infection during the first five days of hospitalization. Performance predictions of blood biomarkers (Serum Amyloid-A (SAA), C-reactive protein, procalcitonin (CRP), white blood cells (WBC), creatinine) and clinical parameters (National Institutes of Health Stroke Scale (NIHSS), age, temperature) for the detection of poststroke infection were evaluated individually using receiver operating characteristics curves. Three machine learning techniques were used for creating panels: Associative Rules Mining, Decision Trees and an internal iterative-threshold based method called PanelomiX. The PanelomiX algorithm showed stable performance when applied to two representative subgroups obtained as splits of the main subgroup. The panel including SAA, WBC and NIHSS had a sensitivity of 97% and a specificity of 45% to identify patients who did not develop an infection. Therefore, it could be used at hospital admission to avoid unnecessary antibiotic (AB) treatment in around half of the patients, and consequently, to reduce AB resistance.

## 1. Introduction

Stroke remains a main cause of disability and the second cause of death worldwide. While initial cerebral infarction is associated with elevated rates of morbidity and mortality, infections occurring during the acute phase of stroke have a major effect on patient’s long-term outcome [1]. Pneumonia and urinary tract infections (UTI) are the most common complications, occurring in 20–60% of stroke survivors, prolonging hospital stays and being responsible of 30% of the poststroke deaths [2,3,4,5]. Multiple and different factors contribute to the development of infections in stroke patients, including age, patients’ comorbidities, stroke severity and stroke induced immune system alteration, among others [6]. Physicians face uncertainty in the early diagnosis of infections caused by inconclusive clinical examination, or due to the lack of specific signs, symptoms and routine laboratory tests. Different blood biomarkers such as procalcitonin (PCT), C-reactive protein (CRP), or white blood cells (WBC) have been suggested as possible candidates to help in predicting/identifying patients at risk for poststroke infections [7,8,9]. Serum Amyloid A (SAA) was recently identified as a new and potentially powerful biomarker for the prediction of poststroke associated infection [10,11,12]. However, to date, none of them has been applied in clinical practice.

In the present study, we postulate that a score system combining SAA and clinical variables could better stratify patients at low risk of infection, consequently improving their hospital management and therapy guidance.

## 2. Materials and Methods

### 2.1. Sample and Patient Descriptions

For the present post hoc analysis, the overall cohort of 283 consecutively enrolled ischemic stroke patients (Clini-calTrials.gov.NCT00390962) with available blood samples was divided in two different subgroups, i.e., the discovery subgroup (40 patients) and the validation subgroup (243 patients). They were hospitalized between November 2006 and November 2007 at the University Hospital of Basel in Switzerland. The quality of blood samples was retested before analyzing the biomarkers of the present study. The local ethics committee approved the study, all included patients or their legal representatives signed an informed consent.

Several statistical and visualization methods can be used to assess potential biases arising from the division into two subgroups. One possible solution is the numerical comparison between the demographic characteristics of the two subgroups in Table 1 and Table 2. However, a faster visual test of similitude between the two subgroups is the t-distributed stochastic neighbor embedding (t-SNE) technique (Figure 1). This data visualization tool provides a global view of how representative is the discovery subgroup with respect to the validation subgroup. The t-SNE technique has already been used in the medical field for dimensionality reduction and for detecting clusters in the data [13,14]. The t-SNE visualization for the present study, shown in Figure 2 below, was obtained using the t-SNE package in R [15] for the entire set of numerical and categorical measurements taken on patients in the two subgroups. By analyzing this graph, we can easily conclude that the information in the two subgroups covers the same areas in the two-dimensional representation, and that there are no visible clusters of patients.

### 2.2. Definition of Stroke Associated Pneumonia

Infections that developed within the first five days of hospitalization were considered Stroke Associated Infections (SAI). According to the U.S. Centers for Disease Control and Prevention (CDC), three types of infections were distinguished: pneumonia, urinary tract infection (UTI) and other infections (OI). Pneumonia was diagnosed when among the two following symptoms groups, at least one symptom of each group was present: (1) abnormal respiratory examination, pulmonary infiltrates in chest X-rays; (2) productive cough with purulent sputum, positive microbiological cultures from the lower respiratory tract or blood cultures. Similarly, for the diagnosis of UTI two of the following symptoms need to be fulfilled: fever (≥38.0 °C), urine sample positive for nitrite, leukocyturia (≥40/μL), or significant bacteriuria (≥104/mL of an uropathogen). Finally, diagnosis of OI was done when white blood cell count was ≥11,000/mL and CRP ≥10 mg/L or temperature was ≥38.0 °C and an infectious manifestation was present.

In order to avoid the inclusion of patients with ongoing infections at admission, patients arriving at the hospital with a temperature > 38 °C, requiring mechanical intubation or patients with an infection prior to the stroke were excluded of the study.

### 2.3. Blood Sample Collection

Blood samples were collected in EDTA by venous puncture and afterwards centrifuged at 3000× *g* during 30 min. Collection of the blood was conducted on admission to the hospital and within the first three days following the stroke onset, as well as one, three and five days after admission. At these time points SAA concentrations were measured according to manufacturer instructions and using the Vascular Injury Panel-II (Meso Scale Discovery, Rockville, Maryland 20850-3173). Plasma samples were diluted 1:1000 using the kit diluent. To evaluate the concentration of each sample an ECL detection system (SECTOR Imager 2400, MesoScale Discovery, Rockville, Maryland 20850-3173) was used.

Levels of CRP, PCT, WBC and monocytes were measured during clinical routine at day 1.

### 2.4. Statistical Analyses

Statistical analyses were carried out using R version 3.6.3 (2020-02-29).

Mann-Whitney test was used to detect statistically significant clinical parameters that would indicate that patients’ measurements in one group (infectious/non-infectious) tend to be significantly larger than patients in the other group (infectious/non-infectious). The Mann–Whitney U-test was used to compare the two unpaired groups. It was also the test used to evaluate the numerical variables: age, biomarkers (SAA, CRP, WBC, Monocytes) or clinical parameters (NIHSS).

Fisher’s exact test (for small counts) and the Chi squared test (for relatively large counts) were used to test associations between presence of infection and categorical variables such as gender, cardiovascular risk factors (CVRF) or TOAST.

The performance of individual biomarkers were evaluated using receiver operating characteristics (ROC) curves with pROC package [16]. Area under the curve (AUC), specificity (SP) and sensitivity (SE) analysis were performed on the different clinical parameters (NIHSS, WBC, monocytes) and biomarkers (SAA, CRP and PCT) to evaluate their capacity in discriminating patients that will or will not develop an infection.

To find the best panel combination, we trained three different machine learning methods: Classification and Association rule mining, an algorithm that mines the dataset for predictive rules and associations. Decision trees, a very-well known algorithm creating splits in levels of the most representative biomarkers such that the final structure could be illustrated as a tree with nodes and branches [17] and PanelomiX, an iterative search method for the combination of biomarkers and numerical clinical parameters [18]. The three methods were trained on the discovery subgroup and their performance was compared in the validation subgroup.

#### 2.4.1. Classification and Association Rule Mining

An association rule is a conditional event that states the occurrence of event B if event A has happened previously. Mainly used in basket analyses, it is a data-mining technique that searches for all possible associations between measurements of the dataset that satisfy some minimum constraints established by the user. It can easily be extended to perform association rule-based classification, which involves finding rules that accurately predict a single target (class) variable. One can compare association rules using different metrics: support, confidence and lift.

The support (A ⟶ B) is the number of patients with both events A and B divided by the total number of patients. Higher support for a rule indicates that it should apply to large number of patients. The support is simply the frequency of events within the whole dataset.

The confidence (A ⟶ B) is the number of patients with both conditions A and B satisfied divided by all patients with condition A. The confidence is the percentage of patients that satisfy the event A which contain also event B.

The lift (A ⟶ B) is the observed support of events A and B divided by the individual support of event A and event B. It is a measure of independence between events A and B: if a rule has lift 1, then the two events are independent; if lift > 1, it indicates to which degree the two events are dependent of each other and when lift < 1, the two events A and B are substitute of each other.

We used the rules CBA package in R to extract the set of rules for targeting infectious/non-infectious patients in the discovery subgroup. The search for all rules used a constraint of the type: support = 1 for the group of infectious patients.

#### 2.4.2. Decision Trees

Decision trees are intuitive and easy to interpret. The Classification Tree algorithm works by first spitting the training set into two subsets using a single predictor and an associated threshold to this predictor. This first predictor and its threshold are chosen by searching through all predictors for the one that produces the purest subset. Complete purity of one branch of the decision tree means that all patients falling in that category belong to the same outcome class. The algorithm continues to split the data by choosing the best predictor/biomarker at each iteration for each branch recursively. It stops when it no longer finds a predictor that reduces impurity, or when any other regularization options established as stopping rules require it to do so. Decision trees make very few assumptions regarding the data, but if they are left unconstrained, they are likely to overfit. Therefore, to avoid this problem, one needs to restrict the freedom of the Decision Tree by modifying one of its hyperparameters related to purity, the maximum predictors used per split, the depth of the tree, etc.

We used the R package rpart and Decision tree trained on the Discovery Set was regularized by setting the minimum number of patients to 4 in each terminal node.

#### 2.4.3. PanelomiX

PanelomiX is an algorithm described elsewhere [19] that can combine any numerical measurement on patients (either clinical parameters or molecule levels) in a multivariate manner. It is an exhaustive search algorithm that selects the optimal mix of biomarkers for the desired specificity or sensitivity by associating each predictor included in a panel with a threshold. It shows results in a sequence of if-else statements very easy to interpret and understand. We used the PanelomiX R package hosted on Github (https://github.com/xrobin/PanelomiX, 01/04/2021).

## 3. Results

### 3.1. Population

The present study included a total of 283 patients, that were divided in two subgroups: the discovery group, with 40 patients (19 non-infected and 21 infected), and the validation subgroup, with 243 patients (204 non-infected and 39 infected). The demographic characteristics of the population are shown in Table 1 (discovery subgroup) and Table 2 (validation subgroup) below.

**Table 1 diagnostics-11-01070-t001:** Demographic characteristics of the discovery subgroup including 19 non-infected and 21 infected patients.

	Discovery Group			
	All Patients	No Infection	Infection	*p* Value
	(*n* = 40)	(*n* = 19)	(*n* = 21)	
Age, median (IQR)	79.2 (70.87–82.02)	78.26 (74.5–80.5)	80.4 (69.5–83)	0.78
Sex, female, n (%)	18	9(50)	9 (50)	1
CVRF, n (%)				
Hypertension	31 (100)	12 (38.7)	19 (61.3)	0.06
Atrial fibrillation	9 (100)	3 (33.3)	6 (66.7)	0.17
Current smoking	11 (100)	5 (45.5)	6 (54.5)	0.64
Diabetes Mellitus	7 (100)	4 (57.1)	3 (42.9)	0.69
Coronary heart disease	10 (100)	4 (40)	6 (60)	0.72
Clinical Scales median (IQR)				
NIHSS	5.5 (2–12)	3 (2–7)	12 (4–14)	0.012
Admission temperature (°C)	37 (36.6–37.2)	37.1 (36.9–37.3)	36.8 (36.2–37.2)	0.093
Biomarkers median (IQR)				
WBC (million/mm^3^)	8.63 (6.97–10)	7.66 (6.3–9.2)	9.3 (7.4–11.2)	0.14
CRP (µg/mL)	3.6 (3–8.95)	3.6 (3–6.65)	4.8 (3–17.4)	0.22
Monocytes (million/mm^3^)	0.41 (0.32–0.52)	0.38 (0.32–0.51)	0.44 (0.32–0.52)	0.63
PCT (ng/mL)	0.017 (0.012–0.025)	0.013 (0.012–0.02)	0.017(0.012–0.028)	0.32
SAA (µg/mL)	8.38 (3.26–21.65)	4.27 (2.32–8.39)	16.8 (7.1–81.4)	0.001
TOAST, n (%)				
Large vessel stroke	8 (100)	3 (37.5)	5 (62.5)	0.369
Cardioembolic	8 (100)	4 (50)	4 (50)	
Microangiopathic	14 (100)	5 (35.7)	9 (64.3)	
Other	0	0 (0)	0 (0)	
Unknown	10 (100)	7 (70)	3 (30)	

Abbreviations: CRP: C-reactive protein, CVRF: cardiovascular risk factors; NIHSS: National Institutes of Health Stroke Scale; PCT: Procalcitonin; SAA: Serum amyloid A; TOAST: Trial of Org 10172 in Acute Stroke Treatment; WBC: White blood count.

**Table 2 diagnostics-11-01070-t002:** Demographic characteristics of the validation subgroup including 204 non-infected and 39 infected patients.

	Validation Group			
	All Patients	No Infection	Infection	*p* Value
	(*n* = 243)	(*n* = 204)	(*n* = 39)	
Age, median (IQR)	74.66 (61.11–81.6)	74 (60.3–81.6)	76.07 (68.7–81.9)	0.26
Sex, female, n (%)	97	72 (74.3)	25 (25.7)	0.001
CVRF, n (%)				
Hypertension	182 (100)	147 (80.8)	35 (19.2)	0.09
Atrial fibrillation	48 (100)	35 (72.9)	13 (27.1)	0.049
Current smoking	93 (100)	75 (80.6)	18 (19.4)	0.59
Diabetes Mellitus	47 (100)	39 (82.9)	9 (19.1)	0.09
Coronary heart disease	51 (100)	46 (90.2)	5 (9.8)	0.21
Clinical Scales median (IQR)				
NIHSS	5 (2.7–3)	4 (2–8)	9 (5.5–18.50)	9.9 × 10^−6^
Admission temperature (°C)	37 (36.6–37.4)	37 (36.6–37.5)	37.1 (36.8–37.3)	0.90
Biomarkers median (IQR)				
WBC (million/mm^3^)	8.03 (6.6–9.8)	7.8 (6.4–9.4)	9.77 (7.75–11.7)	0.0004
CRP (µg/mL)	3 (3–7.1)	3 (3–5.7)	5.6 (3–15.8)	0.0019
Monocytes (million/mm^3^)	0.38 (0.3–0.5)	0.38 (0.29–0.48)	0.38 (0.31–0.51)	0.54
PCT (ng/mL)	0.02 (0.012–0.025)	0.02 (0.012–0.02)	0.02 (0.01–0.03)	0.003
SAA (µg/mL)	5.94 (2.63–19.7)	4.97 (2.52–12.7)	16.7 (4.7–89.2)	0.0007
TOAST, n (%)				
Large vessel stroke	46 (100)	40 (86.9)	6 (13.1)	0.08
Cardioembolic	41 (100)	39 (95.1)	2 (4.9)	
Microangiopathic	83 (100)	63 (75.9)	20 (24.1)	
Other	14 (100)	12 (85.7)	2 (14.3)	
Unknown	58 (100)	49 (84.5)	9 (15.5)	

Abbreviations: CRP: C-reactive protein, CVRF: cardiovascular risk factors; NIHSS: National Institutes of Health Stroke Scale, PCT: Procalcitonin; SAA: Serum amyloid A; TOAST: Trial of Org 10,172 in Acute Stroke Treatment; WBC: White blood count.

In the discovery subgroup, median patient age was 79.2 years old (with interquartile range (IQR): 70.87–82.02). Patients with severe stroke expressed in a higher score of the NIHSS, as well as female patients, were more prone to infections during their hospital stay. Cardiovascular risk factors or the stroke subtype (TOAST) did not significantly affect the risk of infection development

Validation subgroup patients had a median age of 74.66 (IQR: 61.11–81.6), and 60% were men. According to this subpopulation results, the development of a poststroke infection could significantly be affected by gender. Stroke severity (NIHSS) appeared to be an important risk factor. Similarly, as in the discovery subgroup, in this validation subgroup of patients, cardiovascular risk factors, except atrial fibrillation, or the subtype of stroke did not influence the development of an infection.

### 3.2. Infection Prediction According to Clinical Parameters and Blood Biomarkers

When evaluating the overall performance, NIHSS and SAA, with an AUC of approximately 70%, were the two parameters presenting the highest accuracy to differentiate between infectious and non-infectious patients (Table 3). When SE values were fixed to more than 90%, in the discovery subgroup SAA was the biomarker presenting the highest SP value (90% SE, 50% SP). In the validation subgroup however, the most specific biomarker was the NIHSS (92.3% SE, 31.3%SP) followed by SAA (91.6% SE, 26.4% SP) and WBC (92.3% SE, 25.4% SP) (results shown in Table 3).

### 3.3. Infection Prediction According to a Panel of Different Markers

Even if the results are relatively comparable between discovery and validation subgroups, the overall performance of SAA or NIHSS as individual predictors of poststroke infections remains low. Therefore, we have investigated the capacity of three machine learning algorithms (Classification and Associative rules mining, Decision Trees and PanelomiX) to create a set of predictive rules using a selection of biomarkers and clinical parameters. The three machine learning techniques chosen for this prediction exercise are “white box” predictors that use a set of rules that are explicit and simple to categorize patients in two groups: infectious/non-infectious. These supervised learning techniques are trained on the discovery subgroup, resulting in a set of predictive rules that contain a restrained set of biomarkers and clinical parameters. The set of biomarkers selected in the discovery subgroup is tested on the validation subgroup. Because of the small sample size in the discovery subgroup only the selected biomarkers and clinical parameters are retained in the first step, rather than the whole set of rules and cut-off points associated with it. The objective is therefore to extract important biomarkers as members of a panel from the discovery subgroup and estimate cut-off points in the validation subgroup, which is a larger subgroup. Ideally, performance of the panel in the discovery subgroup should be comparable to its performance in the validation subgroup.

#### 3.3.1. Classification and Association Rule Mining

Association rules are rules presenting association or correlation between measurements in the datasets. In the present study, the biomarkers that have the highest values of lift are SAA, NIHSS and WBC, with the interpretation that these three biomarkers contain regions of values that are the most associated with non-infectious patients. The rules in the discovery (Table 4) and validation subgroups (Table 5) are generated using a constraint on the support for the infectious patients (support = 1) Consequently, the combination SAA, NIHSS, WBC has been the panel of biomarkers selected to construct rules on the validation subgroup (Table 5). The values of lift are now higher than 3, indicating a strong association between SAA, NIHSS and WBC for the infectious group of patients. The highest values for confidence are related to rules containing WBC and NIHSS, with the interpretation that 92% of the patients which have an NIHSS < 6.5 did not develop an infection. The performance of the rules containing SAA, NIHSS and WBC in both discovery subgroup and validation subgroup are shown in Table 6 In the discovery subgroup 100% SE was obtained for 63.2% SP while in the validation subgroup the SE decrease to 83% for a SP value of 51%.

#### 3.3.2. Decision Trees

Results of decision trees are shown in Figure 2. Decision trees applied to the discovery subgroup identify that the pair of biomarkers SAA and PCT together with the clinical parameter NIHSS are the most suitable to discriminate between infectious and non-infectious patients. The structure of the decision rules for the discovery subgroup can be seen in Figure 2 (left). A second decision tree with only these three biomarkers was built on the validation set where the regularization parameter of the number of patients in terminal node was set to 9. The resulting tree is shown in Figure 2 (right). The performance of each Decision tree is shown in Table 6. In the discovery subgroup, SAA, PCT and NIHSS combination showed 90.4% for a SP value of 73.6%. In the validation subgroup however, SE reached a value of 46.1% for a SP value of 94.1%.

#### 3.3.3. PanelomiX

Using PanelomiX, the third and the last machine learning method used in this manuscript, we investigated in the discovery subgroup which was the best panel combination for obtaining high specificity or high sensitivity. SAA, NIHSS and WBC, the biomarkers that created the panel, were tested again in the validation subgroup. The results and cutoffs are shown in Table 7. In the discovery subgroup, for a SE value of 100% 61%SP was found while in the validation subgroup SP value decreased until 47% for 97% SE.

The overall results of different methods are summarized in Table 6. PanelomiX results are presented for high SP and SE, because this method makes it possible to customize the search for the most discriminative rules, producing either high sensitivity rules or high specificity rules, while the two other algorithms optimize an overall performance metric. The discovery subgroup is a step which identifies the most promising biomarkers; only these biomarkers are subsequently used to build classification rules on the validation subgroup using the three machine learning methods.

## 4. Discussion

The present study has shown important findings related to poststroke associated pneumonia, one of the most important causes of death and disability after stroke onset. The discovery of a biomarker or panel of biomarkers able to detect patients at low risk of infection development could importantly improve their hospital care and avoid antibiotic therapy. Results of this manuscript show that the panel combination of SAA, NIHSS and WBC obtained with PanelomiX, could be a promising tool to stratify patients at low risk of infection.

More concretely, we have shown that this panel was able to reach SP values of 61% (discovery subgroup) and 41.7% (validation subgroup) when SE values were fixed to more than 90%. According to these results it would be possible to avoid antibiotic treatment on an important number of patients that are at low risk of infection development, improving their hospital management but also decreasing the overuse of antibiotics and reducing the risk of resistance associated to this treatment.

Individually, NIHSS and SAA, with an AUC of nearly 70%, were the two parameters presenting the highest accuracy to differentiate between infectious and non-infectious patients. In the validation subgroup when the SE was blocked to at least 90%, they reached SP values of 31.3% and 26.4% respectively. This means that using them alone, NIHSS could help to identify at admission approximately 1/3 of the patients that are not at risk of infection development and SAA ¼ of them. Clinical severity of patients represented by high values of NIHSS has already been shown to be associated with higher infection rate. Previously published results show that patients with higher stroke severity, lower levels of consciousness suffer more often from dysphagia, aspiration and consequently from infection [19].

SAA, has also been described as a potential biomarker for prediction of poststroke infection [9,10,11]. It is an acute phase inflammatory molecule that acts very fast predicting the patients at risk of infection even before the occurrence of clinical signs. During the last couple of years, findings of animal studies led to the hypothesis that in the acute phase of stroke an immunodepression and anti-inflammatory response occurs which increases the risk of poststroke infection. Also different clinical studies have described this anti-inflammatory response, finding an increase in the number of inflammatory markers and impairments in cell-mediated immunity [20]. In the same way, it has already been shown that patients with higher stroke volume and/or severity had higher WBC count, which correlates perfectly with our results [7].

In the present manuscript, the best combination of biomarkers: SAA, WBC, NIHSS was obtained by comparing three machine learning algorithms. They have been applied in a small discovery subgroup and subsequently validated in a larger validation subgroup.

According to our results, the toolbox PanelomiX has been shown to be the most promising one to select the panel of biomarkers; it is the most stable from the discovery subgroup to the validation subgroup and furthermore it allows calculating panels for either high specificity or high sensitivity according to the clinician need. This flexibility is valuable in medical research because it allows orientating the question in terms of what is the most important group of patients to identify as soon as possible, at hospital admission if necessary. PanelomiX has already led to promising results in the combination of biomarkers for detecting mild traumatic brain injuries [21].

The Classification and Association rules mining is a technique that has been quite intensively exploited in clinical research, such as intensive care management system [17] or smoking cessation pharmacotherapy. An excellent survey of the use of association rule mining in health informatics is provided in previous studies. In the present manuscript results obtained with Classification and Association rules mining are similar to PanelomiX results for high sensitivity. However, it provides rules with quite low performance in the validation subgroup, with sensitivity lower than 90% for example.

In the case of the Decision Tree Algorithm, we have seen that it builds decision rules with the highest overall accuracy. However, it produces results that are not stable from the discovery subgroup to the validation subgroup and it clearly overfits the data in the two subgroups. It is a well-known problem of this class of algorithms. Other problems linked to this method were also highlighted in previous studies [22].

There are other powerful machine learning techniques such as random forests and boosting algorithms that rely on decision trees, but these augmented algorithms no longer preserve the nice interpretability property of basic decision trees. This is the reason they were not used in the present study.

The most important limitation of the present study is the small size of the dataset. We included only samples from 283 patients: 40 for the discovery subgroup and 243 for the verification subgroup. The two subgroups of patients used to discover and verify the panel appear to be relatively comparable according to the demographics of the population. However, there is a cluster of patients that remains with low representation in the discovery cohort.

This limitation as well as the low number of included patients warrant a further validation study to confirm that the obtained panel of molecules can be generalized to a larger group of patients. A prospective clinical study including 1100 stroke patients is ongoing to validate our score and to improve the SP of the panel; according to our results it would be possible to avoid antibiotic treatment on an important number of patients (as SE value of 97% means that we have no false positive cases), however, in order to increase the percentage of patients that benefit from this antibiotherapy stop, values of SP should be increased.

Other important point of the study is the inability of SAA ELISA kits to differentiate among SAA isoforms. In a previous proteomic study, we evaluated the expression of the different isoforms (SAA1 and SAA2) as predictors of infection development [11]. Both isoforms appeared to be significantly higher in infected patients than in non-infected ones. In order, to validate these results in a much larger cohort of patients more specific immunoassays should be developed.

## 5. Conclusions

In our study, the panel SAA, NIHSS and WBC obtained the highest performance in the validation subgroup, for either 95% sensitivity or 95% specificity. The cutoffs for the biomarker SAA, NIHSS and WBC need to be further validated in a larger subgroup.

## Figures and Tables

**Figure 1 diagnostics-11-01070-f001:**
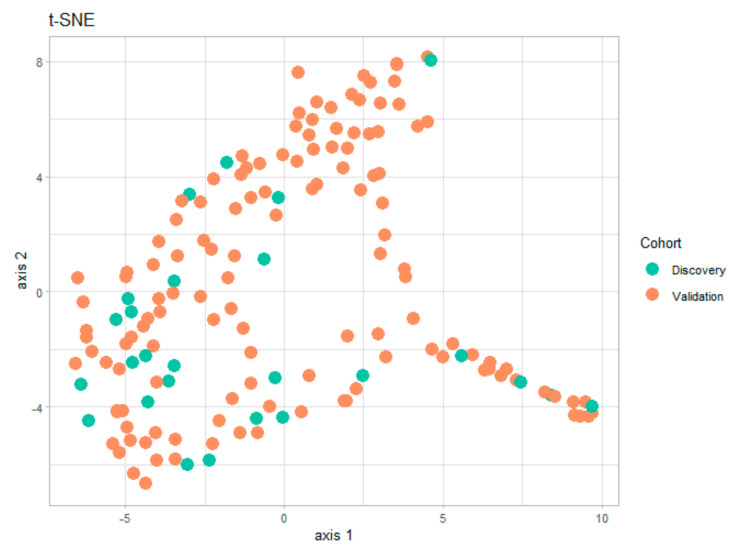
Mapping of all information on the patients in two dimensions using the t-SNE technique. The 2D plot illustrates that the two subgroups cover uniformly the parameter space that there are no visible clusters.

**Figure 2 diagnostics-11-01070-f002:**
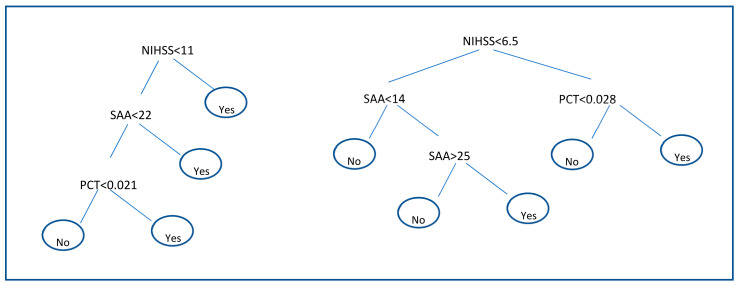
Decision trees applied on the discovery subgroup (left Decision tree) and the Decision Tree applied on the validation subgroup using only biomarkers: SAA and PCT and NIHSS shown on the right.

**Table 3 diagnostics-11-01070-t003:** AUC, SP and SE values of biomarkers in Discovery and Validation Subgroups.

Biomarker	Discovery Subgroup				Validation Subgroup			
	AUC	SE	SP	Cut-off	AUC	SE	SP	Cut-off
SAA (µg/mL)	0.79	90%	50%	3.71	0.68	91.6%	26.4%	2.6
NIHSS	0.73	100%	0%	-	0.72	92.3%	31.3%	2.5
Temperature (°C)	0.66	100%	0%	-	0.50	91.4%	5%	35.9
WBC (million/mm^3^)	0.63	90%	10.5%	5.31	0.68	92.3%	25.4%	6.4
CRP (µg/mL)	0.60	100%	0%	-	0.65	100%	0%	-
PCT (ng/mL)	0.59	90%	15.7%	0.009	0.65	94.6%	8.2%	0.008
Monocytes (million/mm^3^)	0.54	95%	10.5%	0.23	0.53	92.3%	7.6%	0.20

**Table 4 diagnostics-11-01070-t004:** Classification and association rules in the discovery subgroup.

	Left Hand Side	Right Hand Side	Support	Confidence	Lift
1	{WBC = [8.65,9.3)}	{infection = no}	0.100	1	2.1
2	{NIHSS = [1.5,2.5)}	{infection = no}	0.150	1	2.1
3	{SAA = [−Inf,2.56)}	{infection = no}	0.150	1	2.1
5	{SAA = [172, Inf]}	{infection = yes}	0.1	1	1.9
6	{SAA = [21.7,127)}	{infection = yes}	0.1	1	1.9
7	{SAA = [21.7,127)}	{infection = yes}	0.125	1	1.9
8	{MONO = [0.405,0.485)}	{infection = yes}	0.125	1	1.9
9	{MONO = [0.485, Inf], NIHSS = [11, Inf]}	{infection = yes}	0.1	1	1.9
10	{WBC = [9.3, Inf],NIHSS = [11, Inf]}	{infection = yes}	0.15	1	1.9

**Table 5 diagnostics-11-01070-t005:** Classification and association rules in the validation subgroup.

	Left Hand Side	Right Hand Side	Support	Confidence	Lift
1	{WBC = [5.68,6.46}	{infection = no}	0.11	0.96	1.15
2	{NIHSS = [−Inf, 6.5)}	{infection = no}	0.57	0.92	1.09
3	{NIHSS = [6.5,17.5), WBC = [12.2, Inf]}	{infection = yes}	0.02	0.5	3.39
4	{SAA = [147, Inf]}	{infection = yes}	0.03	0.5	3.11
5	{NIHSS = [17.5,26.5)}	{infection = yes}	0.04	0.5	3.11
6	{WBC = [12.2, Inf}	{infection = yes}	0.04	0.5	3.11

**Table 6 diagnostics-11-01070-t006:** Table summarizing performance and biomarkers for the three “white box” algorithm.

		Discovery Subgroup		Validation Subgroup	
Method	Biomarkers	SE	SP	SE	SP
Classification and Association Rules	SAA, WBC and NIHSS	100%	63.15%	83%	51%
Decision Trees	SAA, PCT and NIHSS	90.4%	73.6%	46.1%	94.1%
Panelomix (High SP Panel)	SAA, WBC and NIHSS	63.2%	100%	33.3%	97.1%
Panelomix (High SE Panel)	SAA, WBC and NIHSS	100%	61%	97.2%	47.1%

**Table 7 diagnostics-11-01070-t007:** Performance, Biomarkers and Thresholds for the PanelomiX toolbox.

	Discovery Subgroup			Validation Subgroup		
Label	Biomarkers and Thresholds	SE	SP	Biomarkers and Thresholds	SE	SP
High SP panel	WBC > 11.08 and	63%	100%	WBC > 13.33 and	33%	97%
	SAA > 3.8 and			SAA > 41 and		
	NIHSS > 11			NIHSS > 17.5		
	(infection when ≥2)			(infection when ≥2)		
High SE panel	WBC > 6.5 and	100%	61%	WBC > 10 and	97%	47%
	SAA > 3.8 and			SAA > 10.6 and		
	NIHSS > 7.5			NIHSS > 6.5		
	(infection when ≥2)			(infection when ≥1)		

## Data Availability

Data is contained within the article.

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
