# Peer review of "A Panel Comprising Serum Amyloid A, White Blood Cells and Nihss for the Triage of Patients at Low Risk of Post-Stroke Infection"

_diagnostics, 2021, doi:10.3390/diagnostics11061070_

Round 1
Reviewer 1 Report
SAA ELISA test uses a monocolonal antibody, what if poly clonal antibodies and used- will it provide better specificity and sensitivity? or if the using antibody to SAA1 vs SSA2 peptides make a difference?
Can you elaborate on this aspect in the manuscript.
Author Response
I thank the reviewer for this comment.
In a previous proteomic study published in 2017, we decided to evaluate the trend of the different SAA isoforms. (“Proteomic discovery and verification of Serum Amyloid A, a predictor marker of patients at risk of post-stroke infection: a pilot study”)
PRM analyses were performed on 20 stroke patients in order to evaluate whether either of the acute phase isoforms (SAA1 and SAA2) had a more significant effect on infection and inflammatory processes. The high sequence-similarity between the SAA1 and SAA2 isoforms prevented an evaluation of their effects using classic ELISAs. In that study we measured: FFGHGAEDSLADQAANEWGR peptide (unique to SAA1) and GPGGAWAAEVISNAR peptide (unique to SAA2) across 10 infected and 10 non-infected patients. Both peptides were significantly (p < 0.001) more abundant in infected patients than in non-infected ones.
The following paragraph has been included in the manuscript:
“One limitation of the present study is the inability of SAA ELISA kits to differentiate among SAA isoforms. In a previous proteomic study, we evaluated the expression of the different isoforms (SAA1 and SAA2) as predictors of infection development risks. Both isoforms appeared to be significantly higher in infected patients than in non-infected ones. In order, to validate these results in a much larger cohort of patients more specific immunoassays should be developed.”
Reviewer 2 Report
Interesting and well written article regarding the a "PANEL COMPRISING SERUM AMYLOID A, WHITE BLOOD CELLS AND NIHSS FOR THE TRIAGE OF PATIENTS AT LOW RISK OF POST-STROKE INFECTION"
The idea is interesting, the sensitivity of the panel is very high (97%), while the specificity is low(45%), probably this is the weakest point of the study, this should be better explained in the limitations.
Regarding the references I would suggest to consider these studies regarding laboratory findings and NIHSS supporting your thesis:
- Pilato F, Silva S, et al. Predicting factors of functional outcome in patients with acute ischemic stroke admitted to neuro-intensive care unit—A prospective cohort study. Brain Sciences, 2020, 10(12), pp. 1–14, 911.
- Salsano, G., Pracucci, G, et al.
Complications of mechanical thrombectomy for acute ischemic stroke: Incidence, risk factors, and clinical relevance in the Italian Registry of Endovascular Treatment in acute stroke. International Journal of Stroke, 2020
- Alexandre, A.M., Pedicelli, A,
May endovascular thrombectomy without CT perfusion improve clinical outcome? Clinical Neurology and Neurosurgery, 2020, 198, 106207.
Author Response
Interesting and well written article regarding the a "PANEL COMPRISING SERUM AMYLOID A, WHITE BLOOD CELLS AND NIHSS FOR THE TRIAGE OF PATIENTS AT LOW RISK OF POST-STROKE INFECTION"
The idea is interesting, the sensitivity of the panel is very high (97%), while the specificity is low(45%), probably this is the weakest point of the study, this should be better explained in the limitations.
I completely agree with reviewer comment. In the present article we are mainly interested in stopping antibiotherapy in those patients that are already treated, because suspicion of infection but that are not really infected. This means that with high SE values we avoid false negative cases and we are completely sure that we can stop the treatment in a patient because it is really not infected. Nevertheless, an increase in the SP value would be also interested because it would be traduced by an increase in the number of patients that benefit from this antibiotic treatment stop. “
The following sentence has been included in the manuscript: “more patients and more biomarker combinations should be tested to try to improve the SP of the panel; according to our results it would be possible to avoid antibiotic treatment on an important number of patients (SE value of 97% means that we have no false positive cases), however, in order to increase the percentage of patients that benefit from this antibiotherapy stop, values of SP should be increased.”
Regarding the references I would suggest to consider these studies regarding laboratory findings and NIHSS supporting your thesis:
- Pilato F, Silva S, et al. Predicting factors of functional outcome in patients with acute ischemic stroke admitted to neuro-intensive care unit—A prospective cohort study. Brain Sciences, 2020, 10(12), pp. 1–14, 911.
- Salsano, G., Pracucci, G, et al. Complications of mechanical thrombectomy for acute ischemic stroke: Incidence, risk factors, and clinical relevance in the Italian Registry of Endovascular Treatment in acute stroke. International Journal of Stroke, 2020
- Alexandre, A.M., Pedicelli, A, May endovascular thrombectomy without CT perfusion improve clinical outcome? Clinical Neurology and Neurosurgery, 2020, 198, 106207.
Proposed references have been included in the manuscript
Reviewer 3 Report
This is a thorough piece of work. The authors have developed a score system combining SAA and clinical variables to better stratify post-stroke patients at low risk of infection development. The manuscript is well written and describes well the methodology. However, I believe there is one major point of weakness: the size of the study dataset. This dataset includes only samples from 283 patients that they have divided into a discovery (40 pts) and a validation (243 its) subgroups. These numbers are quite insufficient to affirm the author conclusions.
Although they have made efforts to demonstrate that there were no major laboratory or clinical differences between the two subgroups, this is not clear from the presented data. Indeed, Figure 1 demonstrates that there is a large cluster of patients of the validation set (axis 1 between 0 and 5; axis 2 > 0) that do not share the same parameter space that patients in the discovery set. This raises doubts about the representativity of both populations and the validity of the presented score. The authors should validate this methodology in a new dataset.
Author Response
The reviewer is totally correct about the sample size. However, this manuscript is an exploratory study to evaluate the performance of several machine learning approaches to compare the level of overfitting of each of them.
The following paragraph has been included in the manuscript:
“The most important limitation of the present study is the small size of the dataset. We included only samples from 283 patients: 40 for the discovery subgroup and 243 for the verification subgroup. The two subgroups of patients used to discover and verify the panel appear to be relatively comparable according to the demographics of the population. However, there is a cluster of patients that remains with low representation in the discovery cohort.
This limitation as well as the low number of included patients warrant a further validation study to confirm that the obtained panel of molecules can be generalized to a larger group of patients. A prospective clinical study including 1100 stroke patients is ongoing to validate our score.
